# Genomic Analysis of Two Novel Bacteriophages Infecting *Acinetobacter beijerinckii* and *halotolerans* Species

**DOI:** 10.3390/v15030643

**Published:** 2023-02-28

**Authors:** Marta Gomes, Rita Domingues, Dann Turner, Hugo Oliveira

**Affiliations:** 1Centre of Biological Engineering, University of Minho, 4710-057 Braga, Portugal; 2LABBELS–Associate Laboratory, 4710-057 Braga, Portugal; 3School of Applied Sciences, College of Health, Science and Society, University of the West of England, Bristol Frenchay Campus, Coldharbour Lane, Bristol BS16 1QY, UK

**Keywords:** *Acinetobacter*, bacteriophages, comparative genomics, bioinformatics

## Abstract

Bacteriophages are the most diverse genetic entities on Earth. In this study, two novel bacteriophages, nACB1 (*Podoviridae* morphotype) and nACB2 (*Myoviridae* morphotype), which infect *Acinetobacter beijerinckii* and *Acinetobacter halotolerans*, respectively, were isolated from sewage samples. The genome sequences of nACB1 and nACB2 revealed that their genome sizes were 80,310 bp and 136,560 bp, respectively. Comparative analysis showed that both genomes are novel members of the *Schitoviridae* and the *Ackermannviridae* families, sharing ≤ 40% overall nucleotide identities with any other phages. Interestingly, among other genetic features, nACB1 encoded a very large RNA polymerase, while nACB2 displayed three putative depolymerases (two capsular depolymerases and one capsular esterase) encoded in tandem. This is the first report of phages infecting *A. halotolerans* and *beijerinckii* human pathogenic species. The findings regarding these two phages will allow us to further explore phage—*Acinetobacter* interactions and the genetic evolution for this group of phages.

## 1. Introduction

The *Acinetobacter* genus of the γ-*Proteobacteria* and *Pseudomonadales* order contains more than 30 species of strictly aerobic, non-fermentative, Gram-negative bacilli. The genus includes both non-pathogenic and pathogenic species that are commonly found in soil, water, sewage and foods. In recent years, remarkable advances in genome sequence technologies have revolutionized the field of genomics, allowing for the sequencing of a vast and highly diverse set of genomes at exponentially decreasing costs. As a result, an increasing number of draft or complete genome sequences of *Acinetobacter* spp. have become available [1]. However, they have been focused on members of the *Acinetobacter baumannii-calcoaceticus* (ACB) complex (such as *A. baumannii* and *Acinetobater pittii*), which have been increasingly recognized as important nosocomial pathogens. These species typically colonize patients upon admission to intensive care units (ICU) and rapidly develop resistance to even the last resort antibiotic treatments. Of concern, *A. baumannii* is responsible for up to 6% of nosocomial infections (e.g., ventilator-associated pneumonia, bloodstream, urinary tract and surgical wound infections) but up to 20% of ICU infections worldwide [2]. Moreover, *A. baumannii* was ranked as a major cause of mortality associated with drug-resistant infections in 2019 [3].

Recent genomic studies have mostly analysed the diversity of *A. baumannii* genomes and focused on what distinguishes them from the ACB complex. Although the diversity of the genus remains largely unexplored, several studies have shown that *A. baumannii* gene repertoires are very diverse, with fewer than half of the genes being part of the species’ core-genomes [4].

Similarly, studies of bacteriophages (phages) infecting *Acinetobacter* have been mostly focused on the viruses infecting species of the ACB complex, due to their importance as nosocomial pathogens. Phages are viruses that kill bacteria (natural bacterial predators) and can be used as an alternative therapy to control drug-resistant infections, such as *A. baumannii* [5,6]. Two studies have detailed the comparative analysis of the genomes of complete *Acinetobacter* phages, which expanded in the public databases from 37 (in 2017) [7] to 137 (in 2021) [8]. Both studies demonstrated that the *Acinetobacter* phages displayed vast genomic diversity and evolutionary relationships. The phages infecting *Acinetobacter* sp. can be grouped into eight clusters (subfamilies), forty-six sub-clusters (genera). However, since >98% of the genomes currently deposited in the International Nucleotide Sequence Database Collaboration (INSDC) belong to the phages that infect species of the ACB complex, the genetic reservoir of *Acinetobacter* phages is far from being disclosed.

This study focused on the isolation and analysis of the genome sequences of two novel phages infecting non-ACB complex species, namely *A. beijerinckii* NIPH 838^T^ (phage nACB1) and *A. halotolerans* ANC 5766^T^ (phage nACB2). *A. beijerinckii* and *A. halotolerans* are members of the *Acinetobacter* hemolytic clade [9], which is the largest well-separated phylogroup within the genus [10]. The clade encompasses species that almost all show a strong ability to lyse sheep erythrocytes in ordinary agar media (including both *A. beijerinckii* and *A. halotolerans*) and may include strains with gelatinase activity, whereas species outside this clade are neither hemolytic nor proteolytic, except for a few that embrace a certain proportion of weakly hemolytic strains [9]. Here, we report the morphology and genome sequences of these two newly discovered *Acinetobacter* viral predators.

## 2. Materials and Methods

### 2.1. Bacterial Strains

Strains from six distinct species of *Acientobacter* genus (*A. baumannii, A. pittii, A. halotolerans, A. lwoffii, A. pseudolwoffii* and *A. beijerinckii*), reference laboratory strains or strains from the Laboratory of Bacterial Genetics in Prague, were used (Table 1**)**. All strains were grown aerobically at 37 °C in Trypticase soy agar (TSA) (Oxoid) or Trypticase soy broth (TSB) (Oxoid) medium.

### 2.2. Phage Isolation, Propagation and Purification

The following nomenclature “nACB” was used to identify phages isolated from the “Non-*Acinetobacter calcoaceticus-Acinetobacter baumannii* (ACB) complex” species. Phage nACB1 (infecting *A. beijerinckii*) and nACB2 (infecting *A. halotolerans*) were isolated and purified as previously described [11]. Briefly, sewage samples from wastewater treatment plant sewage samples located in Braga (Portugal), were enriched with an equal volume of double-strength TSB and *A. halotolerans* and *A. beijerinckii* strains. After overnight incubation with agitation, the presence of phages was tested by visual inspection of plaque forming capacity on bacterial lawns. The phage was screened for activity using the spot-on-lawn and soft agar overlay methods. After a 16 h incubation period at 37 °C, clear and turbid areas on the lawns were considered positive for phages. Phage plaques were streak-purified three times to guarantee purity.

A high-titer suspensions of phages were produced through phage infection and multiplication of a liquid culture at a mid-exponential growth phase. After four hours of incubation, the solution was centrifuged (11,000× *g*, 10 min, 4 °C). Clear suspensions were incubated with 1 M NaCl at 37 °C for 1 h, following an incubation with polyethylene glycol (PEG) 6000 (10%, wt/vol) overnight at 4 °C. Next, samples were centrifuged (11,000× *g*, 10 min, 4 °C), suspended in SM buffer (100 mM NaCl, 8 mM MgSO_4_, 50 mM Tris-HCl, pH 7.5), and incubated with chloroform (1:4, vol/vol). The upper part was collected and the phage titer was assessed using a standard double-layer agar method. Purified phage suspensions were filtered through a 0.22 μm Whatman PES membrane and stored at 4 °C.

### 2.3. Lytic Spectra

The lytic spectra of phages nACB1 and nACB2 was assessed using the drop spot test, against the bacterial strains listed in Table 1. Overnight bacterial cultures (~10^8^ CFU/mL) were spread into TSA plates to form lawns using a soft-agar overlay. Next, a 10-µL drop of phage stock was spotted onto the lawns of each strain and incubated for 16 h at 37 °C, to visualise the presence or absence of inhibition areas.

### 2.4. Whole-Genome Sequencing

The genomic DNA of the phages was purified using phenol-chloroform-isoamyl alcohol extraction as described elsewhere [11]. Briefly, phage suspensions (10^8^ PFU/mL) were mixed with RNase and Dnase (Thermo Fisher Scientific) at a final concentration of 5 mg/mL and incubated for 1 h at 37 °C. Next, the phage suspensions were supplemented with EDTA, proteinase K (Thermo Fisher Scientific) and SDS at 20 mM, 100 mg/mL, and 0.5%, being the final concentrations, respectively, and incubated overnight at 55 °C. Phage DNA was then purified by phenol/chloroform extraction, followed by ethanol precipitation. DNA concentration and quality was assessed using 1% of agarose gels and nanodrop quantifications (NanoDrop One, Thermo Fisher Scientific, Waltham, MA, USA).

DNA libraries were constructed using KAPA HyperPlus Kit, analysed by Agilent Bioanalyzer and Qubit measurements, and sequenced in the Illumina MiSeq platform (Illumina Inc., San Diego, CA, USA), using 300 bp paired-end reads (Stabvida, Portugal). Quality-controlled trimmed reads with BBDuk Trimmer plugin in Geneious Prime v.2019 (Biomatters Ltd., Auckland, New Zealand) were *de novo* assembled into a single contig using Geneious assembler with medium sensitivity parameter. Next, the reads were mapped to the assembled contig to assess coverage. The genomes were reoriented to start according to the most similar reference phages deposited in GenBank, using the progressive Mauve tool [12] or when the defined DNA terminus was identified using PhageTerm on a Galaxy-based server (https://galaxy.pasteur.fr, accessed on 12 January 2023) [13].

### 2.5. Genome Annotation and Pairwise Comparative Analysis

Open reading frames (ORFs) were predicted using Rast [14] and manually inspected. A search for putative tRNA-encoding genes was performed with tRNAscan-SE v.2.0 [15]. The function of protein-encoding genes was searched using BLASTp and HHpred, using default parameters (E value of ≤10^−5^). Putative transmembrane domains were assigned using the Phobius [16], TMHMM v2.0 [17] or HMMTOP [18] software. Signal peptides were predicted by the SignalP v5.0 algorithm [19]. Protein pI and molecular mass were calculated using the Geneious Prime in-built tools. For regulatory elements, putative promoters were identified using PhagePromoter (score above 0.8) [20], while putative rho-independent terminators were identified using ARNold (score below −8) [21]. Moreover, machine learning software PhageAI (https://app.phage.ai/, accessed on 12 January 2023), was used to classify the phage lifecycle as lytic or lysogenic. The detail description of the functional parts of putative genes of the phages nACB1 and nACB2 are presented in Appendix A, respectively.

Pairwise whole-genome nucleotide comparisons were made with BLASTn and visualized in Easyfig v 2.2.3 [22]. For the identification of orthologous groups of proteins, sequences from nACB1 and nACB2 were added to an existing dataset comprising of 134 *Acinetobacter* phages [8]. Protein sequences were clustered using PIRATE [23] with parameters of 30, 35, 40% minimum sequence identity and ≥50% coverage of high-scoring pairs. The obtained pan-genome was converted to a binary matrix and subsequently Jaccard distances were calculated between each pair of phage genomes before hierarchical clustering with the complete linkage method, using the hclust and complexHeatmap functions in R [24]. The hierarchically clustered dendrogram of Jaccard distances was exported for visualisation and annotation using iTOL [25].

### 2.6. Phylogenetic Analysis

Phylogenetic analysis of the large terminase subunit was performed using translated amino acid sequences of phages classified as members of the *Ackermannviridae* or *Schitoviridae*. Sequences were aligned using Clustal Omega and the maximum likelihood phylogenetic trees were calculated using IQTree2 [26], with 1000 ultrafast bootstrap (UFBoot) replicates, SH-ALrt test and best fit substitution model as determined by ModelFinder [27,28].

In addition, to calculate the inter-genomic nucleotide sequence similarities, *Ackermannviridae* or *Schitoviridae* viral genomes were analysed and plotted with the Virus Inter-genomic Distance Calculator (VIRIDIC) using default parameters [29].

### 2.7. Transmission Electron Microscopy

Phage lysates (1 mL, >10^8^ PFU/mL) were centrifuged (25,000× *g*, 4 °C, 1 h) in a Beckman J2-21 centrifuge with a JA-18.1 fixed rotor and suspended twice in an equal volume of tap water. Next, suspensions were deposited on carbon-coated copper nickel grids for 2 min and stained with 2% uranyl acetate (pH 4) (Agar Scientific) for 30 s. Finally, transmission electron micrographs of phages were captured using a JEM-1400 microscope (JEOL, Tokyo, Japan). Individual measurements of five viral particles were taken and averaged using ImageJ [30].

### 2.8. Accession Numbers

The genomic sequences of the *A. beijerinckii* phage nACB1 and *A. halotolerans* phage nACB2 were deposited in the NCBI GenBank database with the accession numbers OQ032511 and OQ032512, respectively.

## 3. Results and Discussion

### 3.1. Morphology and General Genomic Features of Phages nACB1 and nACB2

Previous analysis of *Acinetobacter* phage genomes has been limited to viruses isolated mostly against *A. baumannii* and *A. pittii* [8]. Currently, their genomes range from 31 to 378 kb, are genetically diverse and can be grouped into eight clusters (subfamilies) and forty-six sub-clusters (genera). In order to disclose their genetic diversity, we isolated phages against two non-ACB species named nACB1 (isolated from *A. beijerinckii*) and nACB2 (isolated from *A. halotolerans*). These isolation hosts can be considered etiologic agents of infection. *A. beijerinckii* was found in various human specimens, horse airways, hospital environments, soil, or surface water [31]. In contrast, the ecology of *A. halotolerans* remains unknown, as only one type/strain (from soil) is known. The phages were isolated from sewage samples (Braga, Portugal) using an enrichment method and tested against a panel of different *Acinetobacter* species (Table 1).

The phage nACB1 is an *A. beijerinckii*-infecting virus, which was unable to lyse other *Acinetobacter* strains/species within the panel tested. It formed small clear plaques of 1 mm diameter on TSB double-layer agar plates. The TEM images illustrate that nACB1 has a podovirus morphotype, typical of the class *Caudoviricetes* (Figure 1A). Its virion consists of an icosahedral head (65 ± 3 nm vertex to vertex) and a noncontractile tail with short tail fibers (head edge to tail end 7 ± 1 nm in length). The dimensions are relatively higher compared to members of the *Friunavirus* genus (head 40 nm), the most representative group of the *Acinetobacter* podoviruses [5,32], but similar to the recently proposed *Xceevirus* genus, solely represented by *A. baumannii* VB_ApiP_XC38 (head 65 ± 5 nm) [8,32].

The genome of nACB1 was sequenced and was *de novo* assembled with an average coverage of 80×. Its genome had a linear dsDNA molecule with a length of 80,310 bp with 38.7% G + C content and a coding density of 92.1%. According to the results from PhageTerm, 973-bp direct terminal repeats (DTR) were observed. Pairwise genomic comparison showed that nACB1 had 25% overall nucleotide identity (34% coverage × 75% identity) with the closest phage, the *A. pittii* phage VB_ApiP_XC38 (NC_055823) (Figure 1B) that belongs to the *Schitoviridae* family. The nACB1 genome encoded 95 genes situated on both the positive and negative strands and four tRNA (Thr, Tyr, Pro and one undetermined type). The translation of 89 of these proteins started from an ATG codon, five from a GTG, and one from a TTG codon. Of these, only 34 gene products have predicted functions, while the remaining 61 are unknown (of which 22 are novel i.e., without homologies to viral or prokaryotic sequences). Based on the predicted functions, the DNA packaging and structural proteins (e.g., terminase large subunit, major capsid, portal protein), cell lysis (endolysin) or DNA replication, recombination and modification (e.g., RNA polymerase, DNA-binding protein) modules were identified. Regarding the DNA packaging and structural proteins, we identified the large terminase subunit but not the small subunit. Interestingly, there was a very large 3333 aa (i.e., 9999 bp, one-eighth of its genome) RNA polymerase gene, relatively similar to the *Acinetobacter* phages VB_ApiP_XC38 (67% amino acid identity) and Presley (36% amino acid identity). This large virion-encapsidated RNA polymerase gene responsible for host-independent early RNA synthesis during infection, was originally found in *Escherichia* phage N4-like phages, and is considered to be a signature gene of the family *Schitoviridae* [33]. In the cell lysis module, the endolysin (gp41) of the N-acetylmuramidase family, a spanin (gp42) and holin (gp51) similar to *Acinetobacter* phage VB_ApiP_XC38 proteins, were found. Regarding regulatory elements, 21 promoters and 17 rho-independent terminators were identified in the nACB1 genome. Overall, no gene products of antibiotic resistance, toxins, or genes related to lysogeny were found. The presumed lytic lifestyle of nACB1 was confirmed by the PhageAI tool.

It was found that nACB2 infected the sole known isolate of *A. halotolerans* and did not infect the other strains/species tested (Table 1). It exhibited turbid plaques of 1 mm in diameter with expanding halos, which were difficult to see with the naked eye. This phenotype has been previously linked to phages with tail-associated depolymerases, which specifically recognize and degrade extracellular polymers, such as capsules that promote bacterial virulence [5,6,34]. The TEM micrographs showed that nACB2 has a myovirus morphotype (Figure 2A), composed of an icosahedral head (83 ± 6 nm vertex to vertex) with a contractile tail (163 ± 4 by 22 ± 3 nm).

The genome of nACB2 was sequenced and *de novo* assembled with an average coverage of 487×. The nACB2 genome had a linear dsDNA molecule with a size of 136,560 bp, 39.3% G + C content and a coding density of 92.6. No obvious termini were detected using PhageTerm. It encoded 166 genes and three tRNAs (Arg, Asn, IIe), of which 47 gene products have functions either in the DNA packaging and structural proteins, cell lysis or DNA replication, recombination and modification (Figure 2B). No function could be predicted for 114 gene products, of which 39 did not show any homology to sequences in the available databases. The nACB2 genome is predicted to have 86 promoters, and 38 rho-independent terminators. All genes started with the ATG codon, with the exception of four hypothetical and exonuclease genes, which started with GTG or TTG. Phage nACB2 was predicted to undertake a lytic lifestyle according to the PhageAI tool.

According to the ICTV demarcation criteria, nACB2 belongs to the *Ackermannviridae* family. Comparative analysis demonstrated that nACB2 had only 38% overall nucleotide identity (52% coverage × 72.8% identity) with the closest phage, the *A. baumannii* phage SH-Ab 15599 (MH517022), which had similar morphology but relatively smaller head/tails (head of 88 nm; tail of 88 nm) [35]. At least in part, the lack of nucleotide similarity might be explained as an adaptation of the phage to the codon usage of the *A. halotolerans* host. The lytic cassette of nACB2 remains to be fully annotated, as only a lysozyme family gene (gp129) was identified by homology. Surprisingly, nACB2 is predicted to encode three tail-associated depolymerases in tandem (gp164-166). Two have putative capsular depolymerase activities (gp164-165; pectate lyase 3 domains) and one has an esterase activity (gp166; Lipase; SGNH hydrolase domain). While gp164 has no homologies with proteins of other phages, gp165 and gp166 are only relatively similar to phage SH-Ab 15599 proteins (40% average amino acid identity). Recently, it has been shown that 62% of the *Acinetobacter* phages encode depolymerases that recognize *Acinetobacter* capsules as host receptors, and that are exclusively found in smaller phages with genomes up to 90 kb [8]. Moreover, almost all of these phages encode single depolymerases. Therefore, both nACB2 and SH-Ab 15599 seem to be unique *Acinetobacter* phages encoding three distinct putative depolymerases, which likely contribute to increase their host ranges by recognizing multiple host capsules. The fact that a narrow host range of nACB2 was observed, can be explained by the limited number of strains tested. Nevertheless, phages with multiple depolymerases, often encoded by tandem genes, have been described in the family *Ackermannviridae*, which are comprised of phages that infect distant host taxa [36]. The confirmation of the depolymerase activities can only be performed by the cloning of genes and the assessment of recombinantly expressed proteins.

### 3.2. nACB1 and nACB2 Represent Distinct Acinetobacter Phages Isolates

To obtain a higher resolution view of nACB1 and nACB2 diversity within *Acinetobacter* phages, orthologous groups of proteins were identified and added to our recent dataset comprising of 134 phages infecting this bacterial genus [8]. This allowed us to visualize the percentage of shared protein groups (Figure 3) and the presence/absence of protein groups per phage genome (Appendix A). The resulting binary matrix of the presence and absence of protein groups was used to calculate the Jaccard distance and presented as a dendrogram to visualize the relationships among clusters (Figure 4).

As recently shown in 2022, the *Acinetobacter* phage genomes can be grouped into eight clusters (subfamilies) and forty-six sub-clusters (genera), identified from A to N [8]. Here, we showed that the newly isolated nACB1 is positioned within Cluster H (Figure 3 and Figure 4), representing the third phage reported to infect the genus *Acinetobacter* of the family *Schitoviridae*, after Presley and XC38 isolated from the *A. baumannii* and *A. pittii* hosts, respectively [32,37]. It was found that nACB1 shared 62.5% and 33% of its proteins with XC38 and Presley, respectively (Figure 3). While all three phages show limited nucleotide similarity, each encodes for two RNA polymerase genes in addition to a large virion-associated RNA polymerase, which are considered characteristic for this family (Figure 4).

The *A. halotolerans* phage nACB2 belongs to the Cluster N, together with the phage SH-Ab 15599 isolated from a carbapenem-resistant *A. baumannii* host [38]. Prior analysis has demonstrated a relationship between phage SH-Ab 15599 and members of the family *Ackermannviridae* using TBLASTX [8]. The phage nACB2 showed limited nucleotide sequence similarity to SH-Ab 15599 however, 68% of proteins were shared suggesting that these phages could represent a new subfamily (Figure 3). As such, we have chosen to present these phages within a single cluster (Figure 4).

It is interesting to note that there were some proteins shared between nACB1 or nACB2 and other *Acinetobacter* phages of distinct taxonomies and isolated from different host species (Appendix A), which could be interpreted as conserved in these viruses, despite having evolved to infect different *Acinetobacter* host species. For example, nACB2 encodes structural and DNA metabolism proteins that are grouped with proteins found in the *Tawrogvirinae* (T4-like family *Straboviridae*). In contrast, nACB1 showed far fewer examples of proteins shared with other lineages.

Finally, the phylogenetic trees that were constructed from alignments of the large terminase subunit were in accordance with the results from our proteomic tree analysis. The phages nACB1/XC38 and nACB2/SH-Ab 15599 occupy a separate node to the species currently assigned to the *Schitoviridae* and *Ackermannviridae*, respectively, suggesting a more distant relationship (Appendix A). To complement our genomic comparative analysis, all phages from the tree were imported and analysed in VIRIDIC (Appendix A) [39]. Using both BLASTN and VIRIDIC, the two phages fell outside of the thresholds required for their assignment to existing genera (ICTV demarcation criteria of 95% for species and 70% for genus). We therefore identify these phages as representatives of new species, which would also represent new genera. This conclusion is based upon (i) a low degree of sequence identity to the families *Ackermannviridae* and *Schitoviridae*, (ii) pan-genome analysis demonstrating a relationship between the species identified and (iii) maximum likelihood phylogenetic analysis of the Portal protein against formally classified species of these families of *Caudoviricetes*

Overall, the isolation and analysis of non-ACB complex phages (nACB1 and nACB2) serves to underline the current limitations of our knowledge of phages infecting the diverse *Acinetobacter* genus.

## 4. Conclusions

This study represents the first report of phages isolated from the non-ACB species, *A. beijerinckii* and *A. halotolerans*. Based on the evidence presented and the current ICTV demarcation criteria, both phages represent new and distinct isolates of two separate lineages of the class *Caudoviricetes*, and the families *Schitoviridae* and *Ackermannviridae*. This illustrates that the genetic diversity of the *Acinetobacter* viral predators is still to be fully uncovered. Further work is required to assess the potential therapeutic properties of these newly isolated phages against *A. beijerinckii* and *A. halotolerans*.

## Figures and Tables

**Figure 1 viruses-15-00643-f001:**
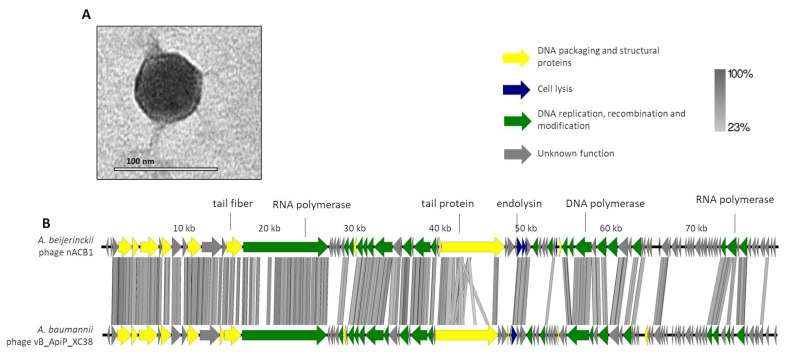
Morphological and genomic analysis of *A. beijerinckii* phage nACB1. (**A**) Transmission electron negatively stained with 2% uranyl acetate. (**B**) Genome map of ACB1 with predicted 95 ORFs numbered and coloured (yellow, green, blue, and gray) according to their predicted functions, and compared using tbBLASTX within EasyFig (Blast minimum length of 100 bp) against the closest phage VB_ApiP_XC38 (NC_055823).A detailed description of the genome can be found in Appendix A.

**Figure 2 viruses-15-00643-f002:**
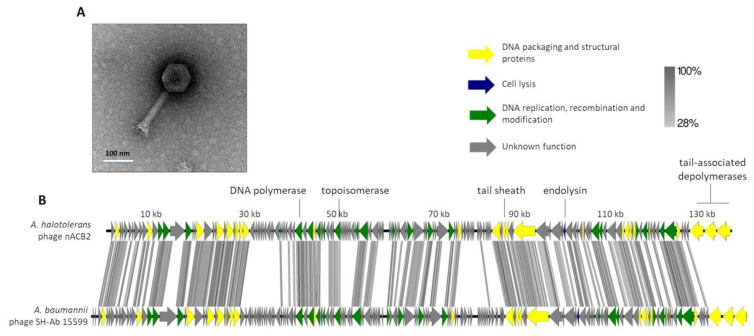
Morphological and genomic analysis of *A. halotolerans* phage nACB2. (**A**) Transmission electron negatively stained with 2% uranyl acetate. (**B**) Genome map of nACB2 with predicted 166 ORFs numbered and coloured (yellow, green, blue, and gray) according to their predicted functions, and compared using tbBLASTX within EasyFig (Blast minimum length of 100 bp) against the closest phage SH-Ab 15599 (MH517022). A detailed description of the genome can be found in Appendix A.

**Figure 3 viruses-15-00643-f003:**
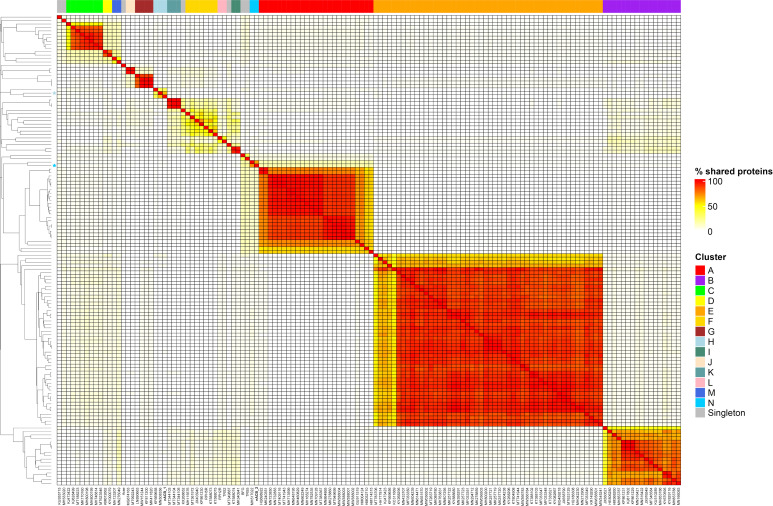
Heatmap of percentage protein groups shared between pairs of *Acinetobacter* phage genomes. The heatmap was produced from the output of the PIRATE pan-genome analysis and hierarchically clustered using the complete linkage method in R. Clusters are indicated by coloured blocks at the top of the heatmap and within the dendrogram according to the key. Coloured bars adjacent to the dendrogram represent clusters described in [8].

**Figure 4 viruses-15-00643-f004:**
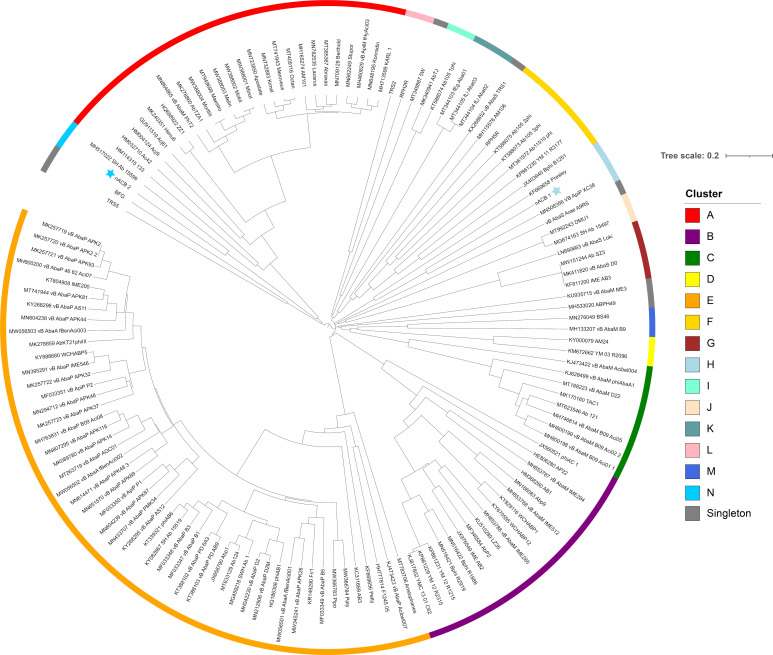
Dendrogram based on hierarchical clustering (complete linkage) of Jaccard distances obtained from the binary matrix of protein presence and absence. The tree is rooted at the midpoint. The scale bar represents the Jaccard distance. Phages nACB1 and nACB2 are indicated by the presence of a star adjacent to the tip label. Coloured bars adjacent to the dendrogram represent clusters described in [8].

**Table 1 viruses-15-00643-t001:** Lytic spectra of the *Acinetobacter* phages nACB1 and nACB2.

Bacteria		Phages	
*Acinetobacter* Strain	Whole-Genome Sequence	nACB1	nACB2
*A. beijerinckii* NIPH 838^T^	APQL00000000.1	+	−
*A. halotolerans* ANC 5766^T^	SGIM00000000.1	−	+
*A. baumannii* NIPH 501^T^	APRG00000000.1	−	−
*A. baumannii* ATCC 17978	NZ_CP018664	−	−
*A. pittii* NIPH 519^T^	APQP00000000.1	−	−
*A. pseudolwoffii* NIPH 5044^T^	PHRG00000000.1	−	−
*A. lwoffii* ANC 512^T^	AYHO00000000.1	−	−

## Data Availability

Not applicable.

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
