# Peer review of "Genomic Analysis of Two Novel Bacteriophages Infecting Acinetobacter beijerinckii and halotolerans Species"

_viruses, 2023, doi:10.3390/v15030643_

Round 1
Reviewer 1 Report
This manuscript presents the genomic analysis of two phages, infecting Acinetobacter beijerinckii and Acinetobacter halotolerans, respectively. That is interesting and could make an important contribution to the further understanding of Acinetobacter phages. However, this manuscript is more like a brief report, and the submission is needed to improve prior to publication. There are some comments as follows,
1. Regarding Figure 1B, please change a better morphological photo of this phage for the well observation.
2. The visualization Software used for figure 1C should be mentioned in this article or drawing statement, and the relative references are required.
3. Line 102, the reference of R software should be added.
4. Regarding phage genome, I would propose a deeper analysis, by using more gene predicting tools (e.g Glimmer version 3.02b, MetaGeneAnnotator, GhostX), before proceeding to the comparison of the results. Also, putative protein functions could be annotated by more tools beyond Blastp (e.g NCBI Conserved Domain Database, Phanotate). This will lead to a safer and more accurate annotation and prediction of genes. Furthermore, web tools, like PhageTerm, can predict phage packaging, leading to a better understanding of the two phages.
5. The authors claimed that they isolated two novel Acinetobacter phages. However, based on the results of this article and ICTV requirement, more evidence is needed to clarify the intensive phylogenetic classification in genus level for new isolated phages.
6. One detailed statistical information table for functional parts of putative genes of the two phages including ORF range, Strand, Start condon, Putative function and Best match, and E-value should be presented in the supplementary materials.
Author Response
Reviewer #1
This manuscript presents the genomic analysis of two phages, infecting Acinetobacter beijerinckii and Acinetobacter halotolerans, respectively. That is interesting and could make an important contribution to the further understanding of Acinetobacter phages. However, this manuscript is more like a brief report, and the submission is needed to improve prior to publication. There are some comments as follows,
Authors: We thank the reviewer for their time taken to review this manuscript and for their useful comments which are addressed in the following responses.
1. Regarding Figure 1B, please change a better morphological photo of this phage for the well observation.
Authors: Done. Revision done accordingly.
2. The visualization Software used for figure 1C should be mentioned in this article or drawing statement, and the relative references are required.
Authors: Done. Revision done accordingly.
3. Line 102, the reference of R software should be added.
Authors: Done. Revision done accordingly. Please see the revised manuscript (line 150)
4. Regarding phage genome, I would propose a deeper analysis, by using more gene predicting tools (e.g Glimmer version 3.02b, MetaGeneAnnotator, GhostX), before proceeding to the comparison of the results. Also, putative protein functions could be annotated by more tools beyond Blastp (e.g NCBI Conserved Domain Database, Phanotate). This will lead to a safer and more accurate annotation and prediction of genes. Furthermore, web tools, like PhageTerm, can predict phage packaging, leading to a better understanding of the two phages.
Authors: Done partially. Revision done accordingly. Several details are now provided to analyze the genomes deeper (e.g. codons, promoters, terminators, phageTerm). As for the use of alternative algorithms to predict ORFs (e.g., Glimmer), we did not observe any differences in the called coding regions. The respective genomes were updated in Genbank with the additional annotations obtained. Please see the revised manuscript (lines e.g. 132-141).
5. The authors claimed that they isolated two novel Acinetobacter phages. However, based on the results of this article and ICTV requirement, more evidence is needed to clarify the intensive phylogenetic classification in genus level for new isolated phages.
Authors: Done. We have provided complementary evidence using VIRIDIC to support our claims. Please see the revised manuscript (lines 160-162, 349-358).
The demarcation criteria specified by the Bacterial Viruses Subcommittee of the ICTV at the level of genus and species is based on whole genome similarity. Using both BLASTN and VIRIDIC, the two phages fall outside of the thresholds required for their assignment to existing genera. We believe that the evidence presented is sufficient to identify these phages as representatives of new species, and which would represent new genera. This conclusion is based upon (i) a low degree of sequence identity to the families Ackermannviridae and Schitoviridae, (ii) pan-genome analysis demonstrating a relationship between the species identified and (iii) maximum likelihood phylogenetic analysis of the Portal protein against formally classified species of these families of Caudoviricetes.
6. One detailed statistical information table for functional parts of putative genes of the two phages including ORF range, Strand, Start condon, Putative function and Best match, and E-value should be presented in the supplementary materials.
Authors: Done. Revision done accordingly. Please see new Tables S1 and S2.
Reviewer 2 Report
The manuscript under review is devoted to the study of the genomes of two new bacteriophages infecting bacteria of the Acinetobacter genus.
The article made a very good impression as a coherent and complete work, which is compiled according to a clear and logical plan. The conducted studies are based on modern methods of molecular biology with a comprehensive bioinformatics processing of the results. Of particular interest are the results of a broad comparison of the genomes of new phages with the genomes of other phages of bacteria of the Acinetobacter genus. There are minor corrections that will make the article better:
1. It is necessary to supplement the introduction with brief information about what diseases A. baumannii causes in humans, what organs or organ systems it affects, and what it leads to.
2. In line 44, the sentense - "have emerged as an alternative therapy to control drug-resistant infections" should be replace by - "can be used as an alternative therapy to control drug-resistant infections".
3. Under reference number 9 (line 75), it would be more correct to refer to the article - “Oliveira H, Pinto G, Oliveira A, Oliveira C, Faustino MA, Briers Y, Domingues L, Azeredo J. Characterization and genome sequencing of a Citrobacter freundii phage CfP1 harboring a lysin active against multidrug-resistant isolates. Appl Microbiol Biotechnol. 2016 Dec;100(24):10543-10553. doi: 10.1007/s00253-016-7858-0” where the method of phage isolation and purification is described in detail.
4. Method 2.6 -Transmission Electron microscopy should describe in more detail the procedure for centrifugation of the phage lysates.
Author Response
Reviewer #2
The manuscript under review is devoted to the study of the genomes of two new bacteriophages infecting bacteria of the Acinetobacter genus.
The article made a very good impression as a coherent and complete work, which is compiled according to a clear and logical plan. The conducted studies are based on modern methods of molecular biology with a comprehensive bioinformatics processing of the results. Of particular interest are the results of a broad comparison of the genomes of new phages with the genomes of other phages of bacteria of the Acinetobacter genus. There are minor corrections that will make the article better:
We thank the reviewer for their time taken to review this manuscript and for their useful comments which are addressed in the following responses.
1. It is necessary to supplement the introduction with brief information about what diseases A. baumannii causes in humans, what organs or organ systems it affects, and what it leads to.
Authors: Done. Revision done accordingly. Please see the revised manuscript (lines 39-43)
2. In line 44, the sentence - "have emerged as an alternative therapy to control drug-resistant infections" should be replace by - "can be used as an alternative therapy to control drug-resistant infections".
Authors: Done. Revision done accordingly. Please see the revised manuscript (line 52)
3. Under reference number 9 (line 75), it would be more correct to refer to the article - “Oliveira H, Pinto G, Oliveira A, Oliveira C, Faustino MA, Briers Y, Domingues L, Azeredo J. Characterization and genome sequencing of a Citrobacter freundii phage CfP1 harboring a lysin active against multidrug-resistant isolates. Appl Microbiol Biotechnol. 2016 Dec;100(24):10543-10553. doi: 10.1007/s00253-016-7858-0” where the method of phage isolation and purification is described in detail.
Authors: Done. Revision done accordingly. Still, we detail more the descriptions. Please see the revised manuscript (lines 92-100)
4. Method 2.6 -Transmission Electron microscopy should describe in more detail the procedure for centrifugation of the phage lysates.
Authors: Done. Revision done accordingly. Please see the revised manuscript (lines 165-171)
Reviewer 3 Report
Gomes et al. have provided genomic and phylogenetic analyses of two novel phages infecting Acinetobacter beijerinckii and halotolerans. However, the manuscript is well written and could shed some light on the unknown diversity of Acinetobacter phages there are a few concerns that should be addressed before the manuscript is ready for publication:
line 87: Since the assembly step is critical for phage genomics, I suggest the authors use a better assembler, SPAdes, in this step
line 92: I recommend the authors compare their manual annotation with an annotation pipeline, i.e., RAST
line 116: 4% UA might be a typo here, as it is uncommon to use such a high concentration for TEM
Figure 1, panel b: I recommend authors change this image since neither the icosahedron shape of the capsid nor the phage tail is observable
Figure 2, panel a: the plate image is not helpful as phage plaques are not visible. This is also an issue with the plate picture in figure 1.
Author Response
Reviewer #3
Gomes et al. have provided genomic and phylogenetic analyses of two novel phages infecting Acinetobacter beijerinckii and halotolerans. However, the manuscript is well written and could shed some light on the unknown diversity of Acinetobacter phages there are a few concerns that should be addressed before the manuscript is ready for publication:
We thank the reviewer for their time taken to review this manuscript and for their useful comments which are addressed in the following responses.
line 87: Since the assembly step is critical for phage genomics, I suggest the authors use a better assembler, SPAdes, in this step
Authors: Done. We have performed assembly using SPAdes, however we noted that the results were in agreement with our previous assemblies.
line 92: I recommend the authors compare their manual annotation with an annotation pipeline, i.e., RAST
Authors: Done. Revision done accordingly.
line 116: 4% UA might be a typo here, as it is uncommon to use such a high concentration for TEM
Authors: Done. Indeed a typo. It was 2%. Please see the revised manuscript (line 168)
Figure 1, panel b: I recommend authors change this image since neither the icosahedron shape of the capsid nor the phage tail is observable
Authors: Done. Revision done accordingly.
Figure 2, panel a: the plate image is not helpful as phage plaques are not visible. This is also an issue with the plate picture in figure 1.
Authors: Not Done. We tried several times to improve the quality of the pictures without success. Phage plaques are difficult to see even at naked eye. We have therefore decided to remove them, and keep the descriptions about the morphologies of the phage plaques in the results section.
Reviewer 4 Report
1.VIRIDIC is a new tool to calculate virus intergenomic similarities,it would be much better if the result 3.2 of this manuscript was analysed using this tool.
2.Check Figure 2,two figures in this part were denoted "B" .
Author Response
Reviewer #4
1.VIRIDIC is a new tool to calculate virus intergenomic similarities,it would be much better if the result 3.2 of this manuscript was analysed using this tool.
Authors: Done. Revision done accordingly. Please revised manuscript (lines 160-162, 349-357). We would note that VIRIDIC only allows for the prediction of genus and species level relationships. While low levels of intergenomic similarity may indicate a subfamily level relationship between phage genomes, this needs to be assessed through the identification of orthologous proteins, as has been presented in the manuscript.
2.Check Figure 2, two figures in this part were denoted "B".
Authors: Done. Revision done accordingly.
Round 2
Reviewer 3 Report
The authors have addressed my concerns about their manuscript; therefore, I have no further comments!